# Carcass, Egg Characteristics and Leg Bone Dimensions of Pigeons of Different Origin

**DOI:** 10.3390/ani14101494

**Published:** 2024-05-17

**Authors:** Kamil Stęczny, Dariusz Kokoszyński, Karol Włodarczyk, Henrieta Arpášová, Michalina Gondek, Mohamed Saleh, Marcin Wegner, Kamil Kądziołka

**Affiliations:** 1Department of Animal Sciences, Faculty of Animal Breeding and Biology, Bydgoszcz University of Science and Technology, 85084 Bydgoszcz, Poland; kamil.steczny@o2.pl (K.S.); michalina_gondek@wp.pl (M.G.); marcin.wegner@op.pl (M.W.); kamil.kadziolka@pbs.edu.pl (K.K.); 2Institute of Agricultural and Food Biotechnology—State Research Institute, 02532 Warsaw, Poland; karwlo19@gmail.com; 3Institute of Animal Husbandry, Faculty of Agrobiology and Food Resources, Slovak University of Agriculture, 94976 Nitra, Slovakia; 4Department of Poultry Production, Faculty of Agriculture, Sohag University, Sohag 82524, Egypt; bydg2016@gmail.com

**Keywords:** King pigeon, carrier pigeon, carcass characteristics, eggs characteristics, leg bones

## Abstract

**Simple Summary:**

Pigeons have been kept since ancient times. Currently, keeping pigeons for flight and meat is highly popular. The aim of this study was to compare carrier and King pigeons at the age of 12 months in terms of weight and carcass composition. In addition, due to the small number of available publications, a comparison was made between the pigeon breeds evaluated in terms of femur and tibia dimensions, egg weight and dimensions, parameters of shell and yolk colour, and physical characteristics of egg content. The results of the study indicate a significant effect of pigeon breeds on carcass, egg, and femur and tibia bone traits.

**Abstract:**

In the past, studies have been conducted on the evaluation of meat traits of pigeons, but the knowledge obtained is incomplete and needs to be expanded. The purpose of this study was to obtain information on the weight and proportion of carcass elements, femur and tibia bone dimensions, and egg characteristics of meat of King breed and carrier pigeons. For this study, 16 carcasses of carrier pigeons and 16 carcasses of King pigeons were used, with 8 carcasses of males and 8 carcasses of females of each breed. Additionally, 20 eggs evaluated were from carrier pigeons and 20 eggs from King breed pigeons. The carcasses and eggs were obtained from birds that were 12 months old. The compared pigeon breeds differed (*p* < 0.05) significantly in terms of the weight of the eviscerated carcass with the neck; the content of neck, wings, pectoral and leg muscles in the carcass; as well as in terms of all specified dimensions of tibia and femur length and width. The origin of the pigeons had an effect (*p* < 0.05) on egg weight and dimensions, egg index, and the other studied egg traits, with the exception of eggshell weight and eggshell yellowness, yolk weight, yolk height, yolk diameter, and yolk index. So far, there have been no studies comparing carrier pigeons and King breed pigeons in terms of femur and tibia bone dimensions, morphological composition and egg dimensions, and egg content traits, which adds to the knowledge in this area and indicates the need for continuation in the future.

## 1. Introduction

Pigeons were domesticated more than 3500 years ago as one of the first birds raised by humans [1]. They were used both for religious purposes as well as to carry messages, and later for consumption [2,3]. In Egypt, pigeon meat was a highly prized delicacy [4]. In the 1980s, however, a typically meat variety of pigeon was bred in the United States [5]. Breeding of meat pigeons also developed in other countries, including the European continent and Asia [6,7].

Globally, pigeon meat, depending on the breed, is obtained from young birds, usually those 28–30 days old with a weight of 400–700 g [8,9,10,11]. Currently, many breeds of meat pigeons are known, and those noteworthy are the King, Strasser, Texan, Couchois, Mondain, Lahore, Giant Homer, Polish Lynx, and Wroclaw Meat [3,8].

Compared to broiler chicken carcasses, pigeon carcasses are characterized by a higher proportion of pectoral muscles and a lower proportion of leg muscles [8,12]. Despite its many advantages, pigeon meat is not popular among consumers due to the small number of breeders, lack of availability of slaughterhouses, and poor profitability of production compared to other slaughter birds [11]. Pigeon meat, especially breast muscle, is characterized by a high protein content and a low cholesterol content [11]. Leg meat, on the other hand, contains a lot of fat, which affects its higher caloric content but improves the palatability of the meat [11,13]. The breast and leg muscles are also characterized by a favourable fatty acid profile [14,15].

Along with meat, bird eggs are one of the staple foods consumed by people from different cultures. They contain high-quality protein and essential nutrients [16]. In poultry production, eggshell quality plays an important role in the laying process [17,18,19]. Numerous studies on the suitability of eggs for hatching in various poultry species show that the suitability of hatching eggs is mainly determined by genetic and environmental factors [20,21,22,23,24,25]. The most important characteristics determining the suitability of eggs for hatching are their weight, shape, and shell quality.

The purpose of this study was to compare carrier pigeons and King breed pigeons in terms of carcass weight and composition, femur and tibia bone dimensions, egg weight and dimensions, parameters of shell and yolk colour, and physical characteristics of the egg. Due to the small number of studies on the evaluation of pigeon eggs and morphometric characteristics of the bones of these birds, it is recommended to continue research in this area in the future.

## 2. Materials and Methods

This study was conducted on purchased eviscerated pigeon carcasses obtained from birds maintained in small populations under typical environmental and nutritional conditions for this bird species. According to the breeder, both pigeon genotypes were kept under the same environmental conditions and fed the same pigeon feed mixture throughout the rearing and laying period and had 24 h access to water.

A total of 32 carcasses were purchased for this study: 16 carcasses each from a particular breed, with 8 carcasses of male and 8 carcasses of female carrier pigeons and King breed pigeons at the age of 12 months. After transporting the carcasses to the University’s laboratory, they were subjected to cooling in a refrigerated cabinet for 24 h at 2 °C. After cooling, the carcasses were individually weighed on an electronic scale WLC 6/12/F1/R1 (Radwag, Radom, Poland) with an accuracy of 0.1 g. Whole gutted carcasses with neck were subjected to simplified dissection according to the method developed and reported by Ziołecki and Doruchowski [26]. During this process, the following parts were sequentially extracted from each carcass: abdominal fat, neck without skin, wings with skin, skin with subcutaneous fat from the whole carcass, pectoral muscles (fillets and tenderloins from both sides), leg muscles (all muscles from both thighs and drumsticks), and the remainders of the carcass. The carcass components extracted during dissection were weighed individually using an electronic balance (WLC 6/12/F1/R, Radwag, Radom, Poland), while their content was expressed as a percentage of the weight of the cold gutted carcass with neck.

In the next stage of the study, leg bones were prepared for evaluation. The femur and tibia were extracted, and a large portion of the adjacent tissue was mechanically removed. To remove the residual flesh and connective tissue, the leg bones were boiled for 30 min. Using electronic callipers (Vorel, Toya S.A., Wrocław, Poland); the greatest and medial length; the greatest breadth and depth of the proximal end; the smallest width of the corpus; and the greatest breadth and depth of the distal end of the femur were measured. The following measurements of the tibia were also taken according to method described by den Driesch [27]: the greatest and axial length; the greatest diagonal breadth of the proximal end; the smallest breadth of the corpus; the greatest breadth of the distal end; and the greatest depth of the distal end. 

In this study, we used 20 eggs from carrier pigeons and 20 eggs from King pigeons. Using an electronic scale (WPS 210C Radwag, Radom, Poland), egg weight (g) was determined with an accuracy of 0.01 g. The length and width (long axis and short axis) of the eggs were measured with a stainless, hardened electronic calipers (Vorel, Toya S.A., Wrocław, Poland). The egg shape index (%) was calculated as the ratio of width to length and expressed as a percentage. Formula [28] was used to calculate the eggshell area:(1)$$P_{S}=4.835\;\times \;W^{0.662}$$
where:P_S_—eggshell area (cm^2^);W—egg weight (g).

Shell colour: L* (colour lightness), a* (redness—red colour intensity), and b* (yellowness—yellow colour intensity) was determined using a colorimeter (CR-400, Konica Minolta, Chiyoda-ku, Japan). The egg contents were knocked out on a glass table, and the egg shells were dried at 105 °C in a dryer (SUP 100 M, Poch S.A., Gliwice, Poland). After drying, shell weight (g) was determined on a Radwag WPS 210C balance. With an electronic micrometer screw on the short axis, shell thickness (mm) was measured to the nearest 0.01 mm.

The height of the thick albumen and yolk (mm) were determined on a glass table with a mirror using a QCD device (TSS Ltd., York, UK). Stainless, hardened electronic calipers (Vorel, Toya S.A., Wrocław, Poland) were used to measure the length and width of the thick albumen (mm), as well as the yolk diameter (mm) along the chalaz line. Yolk colour (L*, a*, b*) was determined using a colorimeter (CR-400, Konica Minolta, Chiyoda-ku, Japan). Haugh units (HU) were calculated according to formula [29]:HU = 100 lg (H + 7.7 − 1.7 W^0.37^)

In which:HU—Haugh units;H—height of thick albumen (mm);W—egg weight (g).

Thin albumen, thick albumen, and yolk were separated from each other, and their masses (g) were determined using a WPS 210C electronic balance (Radwag, Radom, Poland). The weight of the albumen (g) was calculated as the sum of the weight of the thin albumen and the thick albumen. The percentages of thin albumen, thick albumen, yolk, and shell were calculated as a ratio to the weight of the fresh egg before evaluation. The yolk index (%) was expressed as the ratio of yolk height to its diameter and the thick albumen index (%) was the ratio of thick albumen height to its diameter.

The collected numerical data were statistically processed. SAS software (SAS Institute, Cary, NC, USA), version 9.4, was used to calculate the mean arithmetic value, standard deviation (sd), and standard error of the mean (total for both pigeon genotypes) for each studied trait [30]. One-way analysis of variance was used to determine the effect of genotype on the characteristics examined. The significance of the differences in the studied traits of the carcass, eggs, and femur and tibia bones of carrier pigeons and King breed pigeons was verified by using the Student’s *t*-test at *p* < 0.05.

## 3. Results

The King pigeons in our study were characterized by a significantly higher carcass weight as well as a higher percentage share of neck and wings in the carcass compared to the carrier pigeons. On the other hand, the carrier pigeons were characterized by a higher percentage of pectoral muscles in the carcass, which may be due to their adaptation to long flights; the percentage of leg muscles was higher in the King pigeons, while a higher percentage of skin with subcutaneous fat in the carcass was characteristic of the carrier pigeons. The abdominal fat and carcass remainders contents varied, but not significantly (*p* > 0.05), between the compared pigeon breeds (Table 1).

In this study, the greater and medial lengths of the femur, greatest breadth of proximal end, greatest depth of proximal end, smallest breadth of the corpus, greatest breadth of distal end, and greatest depth of distal end were found in the King pigeons rather than the carrier pigeons, and these differences were statistically significant (Table 2).

The King breed pigeons were characterized by a significantly (*p* < 0.05) greater greatest length of the tibia compared to the tibia of the carrier pigeons. The axial length as well as greatest diagonal of proximal end were significantly (*p* < 0.05) higher in the King pigeons than in the carrier pigeons. The size of the smallest breadth of the corpus was greater in the King breed pigeons. In addition, the King breed pigeons were also characterized as having the greater greatest breadth of the distal end. In addition, the King breed pigeons were also characterized by a greater greatest depth of the distal end compared to carrier pigeons, and this difference was also a significant (*p* < 0.05) (Table 3).

The eggs of the King pigeons were characterized by a significantly (*p* < 0.05) greater weight compared to those of the carrier pigeons. Both egg length and width were greater in the King pigeons, and these differences were significant (*p* < 0.05). The egg shape index was 71.0 ± 1.6 in the King pigeons compared to the of 74.4 ± 2.0 index of the carrier pigeon eggs, and these differences were significant (*p* < 0.05) (Table 4).

The eggshell weight of the King breed pigeons was higher than that of the carrier pigeons, but the difference was not significant (*p* > 0.05). Significant (*p* < 0.05) differences were recorded for the eggshell contents in the eggs, with 6.02 ± 0.76% recorded for the King pigeons and 7.06 ± 0.39% for the carrier pigeons, respectively. A smaller eggshell thickness (*p* = 0.015) was characteristic of the King pigeons’ eggs compared to those of the carrier pigeons. Eggshell lightness (L*) and eggshell redness (a*) were higher for the carrier pigeon eggs than the King pigeon eggs, and these differences were statistically significant (*p* < 0.05). Eggshell yellowness (b*) was at a comparable level for both of the pigeon breeds tested (*p* = 0.968). The eggshell surface was 39.0 ± 3.4 in the King pigeons compared to the surface of 34.3 ± 1.8 in the carrier pigeon eggs, and these differences were significant (*p* < 0.05), as seen in Table 5.

From the data presented in Table 6, it can be seen that the eggs of the carrier pigeons were characterized by a higher yolk weight compared to the King breed pigeons, but this difference was not significant (*p* = 0.211). A significant (*p* < 0.001) difference was found in terms of the percentage of yolk in the egg, which was higher in the carrier pigeon eggs than in the King breed pigeon eggs. The yolk height, yolk diameter, and yolk index were at similar levels for both the pigeon breeds (*p* > 0.05). However, significant differences (*p* < 0.05) were recorded in yolk lightness (L*), yolk redness (a*), and yolk yellowness (b*). Albumen weight and percent of albumen content of the egg were significantly (*p* < 0.05) higher for the King pigeon eggs compared to the carrier pigeon eggs. Significant differences (*p* < 0.05) were also noted in thick albumen height, thick albumen index, and Haugh units.

## 4. Discussion

In our study, the King pigeons differed from the carrier pigeons in carcass weight and carcass dimensions. These differences were due to selection in the meat direction within King pigeons. King pigeons, compared to carrier pigeons, are characterized by a fast growth rate and high body weight. The King pigeons’ carcasses were characterized by a higher weight compared to the carrier pigeons. A smaller carcass weight was recorded in King breed pigeons by Kokoszyński et al. [3]. A lower percentage of pectoral muscles in the carcasses of both the King pigeons and carrier pigeons was recorded in our study compared to in the study by Kokoszyński et al. [3]. Jiang et al. also reported a higher breast muscle content in 28-day-old King pigeons compared to this study [31]. The percentages of leg muscles in King pigeons and carrier pigeons were 7% and 5.4%, respectively, in their study. Similar values were reported by Kokoszyński et al. [3], ranging from 5.5 to 7.1 percent. Jiang et al. [31] recorded leg muscle percentages in the range of 6.83–7.97 in 28-day-old King pigeons. Miąsko and Łukasiewicz [12] recorded the percentage of pectoral muscles in 28-week-old Couchois × King and Wroclaw Meat × King pigeons, reporting 7.29% and 8.08%, respectively. In this study, the percentages of neck in the King pigeons and carrier pigeons were 3.4% and 2.6%, respectively, but these differences were statistically insignificant. Similar values in carrier pigeons and King breed pigeons were recorded by Kokoszyński et al. [3], ranging from 2.7 to 3.8%. Higher neck percentages were recorded in carrier pigeons by Abdel-Azeem et al. [32], amounting to 4.6–6.6%. In this study, the carrier pigeons had a higher skin with subcutaneous fat content, which is a natural trait that adapts birds to low temperatures during flight. Similar values were reported by Kokoszyński et al. [3], who also report fat as an energy reserve during flight [33]. In our study, the content of abdominal fat in both groups of the birds was comparable, ranging from 0.7% to 0.9%. A higher content of abdominal fat (0.85–1.61%) was found in pigeon carcasses by Kokoszyński et al. [3] and Abdel-Azeem et al. [32].

Leg bones are well developed in pigeons, while chicks acquire the ability to stand as early as about 5 days after hatching [34,35]. The hind limbs of birds are not only responsible for ground movements but also play an important role in flight and also during take-off and landing [36]. In this study, the measurements of the femurs and tibias of the pigeons were determined according to their origin. The skeleton of a bird is a passive component of the locomotor system but also a reservoir of calcium, and it used in the construction of egg shells and also affects the quality of poultry meat produced [37]. Bone marrow in long bones, in turn, is involved in the formation of white and red blood cells [38]. The greater greatest length of the femur and tibia in our study was found in the King pigeons compared to the carrier pigeons. A smaller femur length was reported by Yan and Zhang [35] in 336-day-old King breed pigeons compared to in our study, but the femur length was higher than in the carrier pigeons in this study. A similar study on duck bone dimensions in relation to origin was presented by Włodarczyk et al. [39]. In the available scientific literature, there is a lack of studies comparing the metric characteristics of femur and tibia bones in King pigeons and carrier pigeons, which prevents a discussion in this regard.

Bird eggs provide both essential nutrients for the embryo and also meet the nutritional requirements for humans [40]. The egg weights in this study were 23.4 g and 19.3 g for the King pigeons and the carrier pigeons, respectively. Okoh et al. recorded lower egg weights in local Nigerian pigeons than those presented in our study [41]. Slightly lower values for egg weight, namely 22.57 g for King pigeons, were recorded by Sun et al. [42]. An egg width of 26.69 to 26.93 mm was recorded in local Nigerian pigeons by Okoh et al., and this value is lower than in our study [41]. The eggshell weights of the King breed pigeons and carrier pigeons in this study were 1.41 g and 1.36 g, respectively. Sun et al. found a higher eggshell weight (1.60 g) in King breed pigeons than that in this study [42]. Okoh et al. also reported a higher eggshell weight in their study [41]. The percentages of shell in our study for King pigeons and carrier pigeons were 6.01% and 7.06%, respectively, and were lower than for King pigeons in the study by Sun et al. [42]. In our study, a higher yolk weight was found in the carrier pigeons compared to the King pigeons. A higher yolk weight was found by Sun et al. [42] in a study on King breed pigeons. However, lower contents were recorded by Okoh et al. [41] in local Nigerian pigeons. The percentages of egg yolk content were 17.46% and 22.52% in the King pigeons and carrier pigeons, respectively. A higher yolk percentage than that in our study was reported by Sun et al. in the eggs of King breed pigeons (19.33%) [42]. The albumen weight found in the current study were 17.93 g and 13.57 g for the King pigeons and the carrier pigeons, respectively, and these values are higher for the King pigeons than in the study by Sun et al. [42]. A similar situation occurred in our study in the case of the protein percentage in the eggs, wherein it was 76.52% for the King pigeons and 70.42% for the carrier pigeons; in the study of Sun et al. [42], it was at a level of 73.56% for King pigeons.

## 5. Conclusions

In conclusion, the King pigeons were distinguished by a significantly higher carcass weight than the carrier pigeons, which is consistent with their use as a meat. They were also characterized by a higher percentage of wing and leg muscles in the carcass, as well as a lower percentage of pectoral muscles, which would result from the adaptation of carrier pigeons to long flights. The carrier pigeons were also characterized by a higher percentage of skin and subcutaneous fat, which was due to the adaptation of these birds to low temperatures during flight. The King pigeon eggs had a greater weight, length, and width compared to those of the carrier pigeons. The egg shell weight of the King pigeons was slightly higher, but its percentage in the egg was statistically lower compared to the carrier pigeons which, combined with the higher weight of the King pigeons, may result in egg damage during hatching. The results obtained on carcass composition will allow for comparisons of the suitability the studied pigeon breeds’ carcasses for culinary purposes, while the results of the studies on egg characteristics can be useful in determining the breeding potential of birds. Due to the small amount of provision on the evaluation of pigeon egg quality and of, in particular, morphometric evaluation of the bones of these birds, further exploration of this topic is recommended. The present work provides information on the composition of pigeon carcasses, as well as information on the evaluation of eggs and bones, which may be useful for pigeon meat consumers.

## Figures and Tables

**Table 1 animals-14-01494-t001:** Carcass characteristics of White King and carrier pigeons at the age of 12 months.

Trait	White King	Carrier Pigeon	SEM	*p*-Value
Eviscerated carcass weight (g)	493.1 ^a^ ± 77.6	316.9 ^b^ ± 53.5	19.1	<0.001
Share of neck in carcass (%)	3.4 ^a^ ± 0.7	2.6 ^b^ ± 0.5	0.1	0.024
Share of wings in carcass (%)	17.3 ^a^ ± 0.6	14.8 ^b^ ± 1.2	0.3	<0.001
Share of pectoral muscles in carcass (%)	26.2 ^b^ ± 1.1	28.1 ^a^ ± 1.7	0.3	<0.001
Share of leg muscles in carcass (%)	7.0 ^a^ ± 1.6	5.4 ^b^ ± 0.7	0.3	<0.001
Share of skin with fat in carcass (%)	13.7 ± 4.2	16.2 ± 3.9	0.7	0.086
Share of abdominal fat in carcass (%)	0.7 ± 0.5	0.9 ± 0.6	0.1	0.326
Share of carcass remainders (%)	31.7 ± 4.4	32.0 ± 3.3	0.6	0.811

^a,b^ means with different superscripts are statistically different between breeds (*p* < 0.05). SEM—pooled standard error of the mean, *n* = 16/genotype.

**Table 2 animals-14-01494-t002:** Dimensions of the femurs of White King and carrier pigeons at the age of 12 months.

Trait	White King	Carrier Pigeon	SEM	*p*-Value
Greatest length (mm)	51.8 ^a^ ± 2.6	43.6 ^b^ ± 2.0	0.8	<0.001
Medial length (mm)	48.6 ^a^ ± 2.3	41.7 ^b^ ± 1.9	0.7	<0.001
Greatest breadth of proximal end (mm)	10.3 ^a^ ± 0.5	8.0 ^b^ ± 0.8	0.2	<0.001
Greatest depth of proximal end (mm)	8.0 ^a^ ± 0.7	6.0 ^b^ ± 0.9	0.2	<0.001
Smallest breadth of the corpus (mm)	4.9 ^a^ ± 0.3	3.7 ^b^ ± 0.3	0.1	<0.001
Greatest breadth of distal end (mm)	11.4 ^a^ ± 0.8	9.0 ^b^ ± 0.8	0.2	<0.001
Greatest depth of distal end (mm)	6.8 ^a^ ± 1.3	5.6 ^b^ ± 0.9	0.2	0.005

^a,b^ means with different superscripts are statistically different between breeds (*p* < 0.05). SEM—pooled standard error of the mean, *n* = 16/genotype.

**Table 3 animals-14-01494-t003:** Dimensions of the tibias of White King and carrier pigeons at the age of 12 months.

Trait	White King	Carrier Pigeon	SEM	*p*-Value
Greatest length (mm)	71.0 ^a^ ± 2.2	61.0 ^b^ ± 2.7	0.9	<0.001
Axial length (mm)	67.9 ^a^ ± 2.1	59.0 ^b^ ± 2.6	1.2	<0.001
Greatest diagonal of proximal end (mm)	11.2 ^a^ ± 0.8	8.4 ^b^ ± 1.1	0.2	<0.001
Smallest breadth of the corpus (mm)	4.2 ^a^ ± 0.4	3.3 ^b^ ± 0.3	0.1	<0.001
Greatest breadth of distal end (mm)	8.8 ^a^ ± 0.7	6.8 ^b^ ± 0.5	0.2	<0.001
Greatest depth of distal end (mm)	7.6 ^a^ ± 0.6	5.9 ^b^ ± 0.7	0.1	<0.001

^a,b^ means with different superscripts are statistically different between breeds (*p* < 0.05). SEM—pooled standard error of the mean, *n* = 16/genotype.

**Table 4 animals-14-01494-t004:** Egg weights and dimensions of White King and carrier pigeons at the age of 12 months.

Trait	White King	Carrier Pigeon	SEM	*p*-Value
Egg weight (g)	23.4 ^a^ ± 2.5	19.3 ^b^ ± 1.2	0.5	<0.001
Egg length (mm)	44.2 ^a^ ± 1.8	39.8 ^b^ ±1.2	0.5	<0.001
Egg width (mm)	31.4 ^a^ ± 1.3	29.6 ^b^ ± 0.7	0.3	<0.001
Egg shape index (%)	71.0 ^b^ ± 1.6	74.4 ^a^ ± 2.0	0.5	<0.001

^a,b^ means with different superscripts are statistically different between genotypes (*p* < 0.05). SEM—pooled standard error of the mean, *n* = 20/genotype.

**Table 5 animals-14-01494-t005:** Characteristics of the eggshells of White King and carrier pigeons at the age of 12 months.

Trait	White King	Carrier Pigeon	SEM	*p*-Value
Eggshell weight (g)	1.41 ± 0.21	1.36 ± 0.06	0.1	0.381
Eggshell content in egg (%)	6.02 ^b^ ± 0.76	7.06 ^a^ ± 0.39	0.2	<0.001
Eggshell thickness (mm)	0.177 ^b^ ± 0.02	0.201 ^a^ ± 0.02	0.2	0.015
Eggshell ligthness (L*)	80.45 ^b^ ± 1.98	82.18 ^a^ ± 2.08	0.4	0.027
Eggshell redness (a*)	0.18 ^b^ ± 0.11	0.43 ^a^ ± 0.35	0.1	0.013
Eggshell yellowness (b*)	1.13 ± 0.64	1.11 ± 0.96	0.2	0.968
Eggshell surface (cm^2^)	39.0 ^a^ ± 3.4	34.3 ^b^ ± 1.8	0.5	<0.001

^a,b^ means with different superscripts are statistically different between genotypes (*p* < 0.05). SEM—pooled standard error of the mean, *n* = 20/genotype.

**Table 6 animals-14-01494-t006:** Characteristics of liquid egg content of White King and carrier pigeon eggs at the age of 12 months.

Trait	White King	Carrier Pigeon	SEM	*p*-Value
Yolk weight (g)	4.09 ± 0.61	4.34 ± 0.46	0.1	0.211
Share of yolk in egg (%)	17.46 ^b^ ± 3.10	22.52 ^a^ ± 1.84	0.6	<0.001
Yolk height (mm)	10.22 ± 0.74	9.96 ± 0.49	0.4	0.271
Yolk diameter (mm)	28.93 ± 1.47	29.03 ± 1.17	0.8	0.846
Yolk index (%)	35.32 ± 2.78	34.31 ±1.97	0.4	0.279
Yolk lightness (L*)	46.31 ^a^ ± 2.26	39.19 ^b^ ± 2.26	2.4	<0.001
Yolk redness (a*)	-4.49 ^a^ ± 1.09	0.18 ^b^ ± 0.64	0.1	<0.001
Yolk yellowness (b*)	32.65 ^a^ ± 3.19	7.14 ^b^ ± 1.67	0.2	<0.001
Albumen weight (g)	17.93 ^a^ ± 2.28	13.57 ^b^ ± 0.98	0.5	<0.001
Albumen content in egg (%)	76.52 ^a^ ± 3.37	70.42 ^b^ ± 1.84	0.7	<0.001
Thick albumen height (mm)	4.76 ^a^ ± 0.62	3.63 ^b^ ± 0.64	0.1	<0.001
Thick albumen index	0.08 ^a^ ± 0.01	0.07 ^b^ ± 0.01	0.1	<0.001
Haugh units (%)	84.52 ^a^ ± 5.36	79.57 ^b^ ± 4.33	1.0	0.012

^a,b^ means with different superscripts are statistically different between genotypes (*p* < 0.05). SEM—pooled standard error of the mean, *n* = 20/genotype.

## Data Availability

Data are contained within the article.

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
