# Peer review of "Carcass, Egg Characteristics and Leg Bone Dimensions of Pigeons of Different Origin"

_animals, 2024, doi:10.3390/ani14101494_

Round 1
Reviewer 1 Report
Comments and Suggestions for Authors
Dear Authors, your Manuscript is very well written. The study is well designed and executed and merits publication therefore I recommend Minor revision.
I suggest the title change, to make it more attractive and interesting for the reader. My suggestion is as follows, however the Authors should decide.
„Species-specific leg dimensions, carcass and egg characteristics in King and Carrier pigeon“
Lines 74-77 put this sentence into the Conclusion
Line 121 Put space between 105 and °C and check this throughout the text, be consistent, and watch out for small language mistakes.
Line 159 „age“
Line 286-288 Put this sentence at the beginning of the Conclusion paragraph
General: why use SEM and not SD? I suggest the authors present the data in graphs so it is clearer for the reader
Author Response
Comments made by Reviewer 1 and response to the comments.
Comment 1
I suggest the title change, to make it more attractive and interesting for the reader. My suggestion is as follows, however the Authors should decide.
„Species-specific leg dimensions, carcass and egg characteristics in King and Carrier pigeon“
Answer
The title has been changes in line with Reviewer’s 3 proposal. “Carcass, egg characteristics and leg bone dimensions of pigeons of different origin”
Comment 2
Lines 74-77 put this sentence into the Conclusion
Answer
The sentence has been moved to the Conclusions chapter.
Comment3
Line 121 Put space between 105 and °C and check this throughout the text, be consistent, and watch out for small language mistakes.
Answer
Has been changed in the whole text
Comment 4
Line 159 „age“
Answer
Corrected
Comment 5
Line 286-288 Put this sentence at the beginning of the Conclusion paragraph
We believe that the text from lines 286-288 is in the right place and it oughn’t be moved.
Comment 6
General: why use SEM and not SD? I suggest the authors present the data in graphs so it is clearer for the reader
Answers
The other two reviewers raised no objections, and the tabulations presented were found to be correct. The standard error of the mean (SEM) provides an estimate of how much the sample mean differs from the population mean, while the SD is the standard deviation, which is a measure of the dispersion of the results around the arithmetic mean.
Reviewer 2 Report
Comments and Suggestions for Authors
In this study, carcass and egg characteristics, as well as leg bone dimensions were compared between carrier pigeon and King breeds, and the results could provide some basic data for pigeon research. However, as the authors point out, the main use of carrier pigeons is competition and messaging, while the king pigeon is a typical meat pigeon breed, and the difference between the two is obvious. Questions and recommendations are as follows:
1. Why did you only choose two different kinds of pigeon varieties for comparison instead of choosing more breeds.
2. In materials and methods, why did you use the purchased eviscerated pigeon carcasses for the experiment instead of raising and slaughtering pigeons by yourselves?
3. In materials and methods, it was not presented whether the two breeds of pigeons were reared in the same way and used the same diet.
4. Although the statistical analysis method used in this paper cannot be said to be wrong, it is better to regard gender as an interference factor and use a two-factor ANOVA method to analyze.
5 As is known, the egg yolk color is closely related to the diet composition, it should be clearly state whether the composition of the diet is the same for both pigeons in materials and methods.
6. in line 147, “coefficient of variation for each studied trait” were calculated, but it is not presented in the results.
Author Response
Comments made by Reviewer 2 and response to the comments.
Comment 1
.Why did you only choose two different kinds of pigeon varieties for comparison instead of choosing more breeds.
Answer
Only 2 pigeon genotypes were selected because King breed pigeons are typical meat pigeons most frequently kept in Poland, while racing pigeons are usually used culinarily after depletion. Other species of racing pigeons are mostly not used for consumption in Poland. Numerous breeders keep ornamental pigeons that are not used for culinary purposes.
Comment 2
In materials and methods, why did you use the purchased eviscerated pigeon carcasses for the experiment instead of raising and slaughtering pigeons by yourselves?
Answer
The Department of Animal Breeding and Nutrition at the Bydgoszcz University of Technology does not have a pigeon farm or a suitable infrastructure for this purpose and, therefore, the capacity to rear pigeons for research purposes. For this reason, pigeon carcasses and eggs were purchased for the study.
Comment 3
In materials and methods, it was not presented whether the two breeds of pigeons were reared in the same way and used the same diet.
Answer
L: 81-83
In the Materials and Methods section, the following information is given: 'According to the breeder, both pigeon genotypes were kept under the same environmental conditions and fed the same pigeon feed mixture throughout the rearing and laying period and had 24-hour access to water.
Comment 4
Although the statistical analysis method used in this paper cannot be said to be wrong, it is better to regard gender as an interference factor and use a two-factor ANOVA method to analyze.
Answer
The presentation of results on the effect of sex was dropped due to the lack of significant differences between male and female leg bone dimensions and carcass characteristics, with the exception of the percentage of abdominal fat, which was significantly higher in females than in males.
Comment 5
As is known, the egg yolk color is closely related to the diet composition, it should be clearly state whether the composition of the diet is the same for both pigeons in materials and methods.
Answer
The information about diet has been added. (COMMENT 3)
Comment 6
.in line 147, “coefficient of variation for each studied trait” were calculated, but it is not presented in the results.
Answer
L146-147
“.. calculate the mean arithmetic value and coefficient of variation.. changed to
‘…“.. calculate the mean arithmetic value, standard deviation (sd), and standard error of the mean (SEM; total for both genotypes of pigeons)
Reviewer 3 Report
Comments and Suggestions for Authors
Title: Carcass and egg characteristics, and leg bone dimensions of pigeons of different origin
Manuscript ID: 2994106
Comments to the author:
Summary: This manuscript explore the carcass, egg characteristics and leg bone dimensions of pigeons of different origin. This sort of studies should always be encouraged as the findings may be adding new information to the poultry industry and strengthen the available literature volume. The approach used by the authors to meet the stated objectives of the study is well explained and will be useful to the industry.
General comments: Title reflects the study carried out. Summary and abstract adequately describe the study conducted. More attention should be put when describing results and their significance. The references used in the manuscript are adequate and the most are recent.
Specific comments: LN: Line Number
LN2-3: Propose to revise the title as: Carcass, egg characteristics and leg bone dimensions of pigeons of different origin
LN31: Please delete the word: statistically. The level of significance must be indicated in parenthesis as: (p<0.05)
LN34: Please delete the word: a significant. Instead keep: an effect.... The level of significance must be indicated in parenthesis as: (p<0.05).
LN49: Please delete the word ''In the world''. Instead use: Globally....
LN57-58: Please provide a valid reference.
LN85: Please correct as: 2℃
LN94: Please provide the manufacturer details as well.
LN99: Please provide the model number, manufacturer details in parenthesis.
LN108: Please provide model number and manufacturer details in parenthesis.
LN110: Please provide a valid reference to this formula.
LN114=115: Units of eggshell surface and egg weight must be presented in parenthesis.
LN122: Please provide the model number, manufacturer details in parenthesis.
LN123: Please provide the model number, manufacturer details in parenthesis.
LN129: Please provide the model number, manufacturer details in parenthesis.
LN135-136: Units of H and W must be present in parenthesis.
LN139: Please provide the model number, manufacturer details in parenthesis.
LN145: Please mention your test. e.g. One -way ANOVA
LN150: Better ovoid the words: ''statistically significant'' appearing throughout the result section. You can present your results with these two words. Pl. see LN 151.
LN151: Please delete.
LN157: Please correct as: not statistically significant...Please indicate probability in parenthesis as: (P>0.05)
Table 1-6: SEM in footnotes should be described as pooled standard error of mean.
Table 1-6: Please indicate the number of samples (n) analysed in footnotes.
LN167: Please indicate the P value in parenthesis (p<0.05)
LN169: Should be revised as: statistically significant....Please indicate the P value in parenthesis (p<0.05)
LN173: Reduce the space.
LN174: Please indicate the P value in parenthesis (p<0.05)
LN177: Please indicate the P value in parenthesis (p<0.05)
LN179: Please indicate the P value in parenthesis (p<0.05)
LN180-181: Please rephrase this sentence and indicate the P value in parenthesis (p<0.05)
LN184-190: Please indicate the P values in parenthesis where relevant.
LN198-207: Please present P values in parenthesis where relevant.
LN205: Please delete.
LN227: Please mention as: 4.6-6.6%.
LN274: Please delete.
LN282: Please delete.
LN315: Please correct as: carrier pigeon...
LN328: Please revise as: pigeon...
LN40: Title should be in sentence case. Scientific names must be kept italic.

Fine. But, moderate revision is required especially in the Results section.
Author Response
Comments made by Reviewer 3 and response to the comments.
Comment 1
LN2-3: Propose to revise the title as: Carcass, egg characteristics and leg bone dimensions of pigeons of different origin
Answer
The title of the article has been changed according to a comment.
Comment 2
LN31: Please delete the word: statistically. The level of significance must be indicated in parenthesis as: (p<0.05)
Answer
A word ‘statistically’ has been deleted, added (p < 0.05)
Comment 3
LN34: Please delete the word: a significant. Instead keep: an effect.... The level of significance must be indicated in parenthesis as: (p<0.05).
Answer
Corrected
Comment 4
LN49: Please delete the word ''In the world''. Instead use: Globally....
Answer
Corrected
Comment 5
LN57-58: Please provide a valid reference.
Answer
Added
Comment 6
LN85: Please correct as: 2℃
Answer
According to instructions for authors to animals journal, the correct form of writing is a space between the number and the unit.
Comment 7
LN94: Please provide the manufacturer details as well.
Answer
WLC 6/12/F1/R changed to:(WLC 6/12/F1/R, Radwag, Radom, Poland),
Comment 8
LN99: Please provide the model number, manufacturer details in parenthesis.
Answer
L:102
Added „(Vorel, Toya S.A. WrocÅ‚aw Poland),
Comment 9
LN108: Please provide model number and manufacturer details in parenthesis.
„…a Stainless Hardened electronic calliper” changed to
A Stainless Hardened electronic calliper „(Vorel, Toya S.A. WrocÅ‚aw Poland),
Comment 10
LN110: Please provide a valid reference to this formula.
Answer
This is the relevant References. Subsequent articles cite this source.
Comment 11
LN114=115: Units of eggshell surface and egg weight must be presented in parenthesis.
Answer
Measurement units added - cm2 and g.
Data for eggshell surface has been added to Table 5
Comment 12
LN122: Please provide the model number, manufacturer details in parenthesis.
Answer
a colorimeter (Konica Minolta, Chiyoda-ku, Japan) with a CR-400 head changed to a colorimeter (CR-400, Konica Minolta, Chiyoda-ku, Japan)
Comment 13
LN123: Please provide the model number, manufacturer details in parenthesis.
dried at 105°C in a SUP 100 M type dryer (Poch S.A., Gliwice, Poland) changed to dried at 105°C in a dryer (SUP 100 M, Poch S.A., Gliwice, Poland)
Comment 14
LN129: Please provide the model number, manufacturer details in parenthesis.
Answer
Changed to:
“…..using a colorimeter (CR-400, Konica Minolta, Chiyoda-ku, Japan)
Comment 15
LN135-136: Units of H and W must be present in parenthesis.
Answer
Added units for H - height of thick albumen - (mm) and for W - egg weight (g)
Data for Haugh units has been added to Table 6
Comment 16
LN139: Please provide the model number, manufacturer details in parenthesis.
Answer
“...a Radwag WPS 210C balance changed to:
“....a WPS 210C electronic balance (Radwag, Radom, Poland)
Comment 17
LN145: Please mention your test. e.g. One -way ANOVA
Answer
Added:
One-way analysis of variance was used to determine the effect of genotype on the characteristics examined.
Comment 18
LN150: Better ovoid the words: ''statistically significant'' appearing throughout the result section. You can present your results with these two words. Pl. see LN 151.
Answer
Deleted “statistically” (L151, 157, 169, 173, 179, 181, 185, 210)
Comment 19
LN151: Please delete.
Answer
L151
Deleted “statistically”
Comment 20
LN157: Please correct as: not statistically significant...Please indicate probability in parenthesis as: (P>0.05)
Answer
Changed to „not significantly (p > 0.05)”
Comment 21
Table 1-6: SEM in footnotes should be described as pooled standard error of mean.
Answer
In footnotes added:
SEM - pooled standard error of mean
Under Tables 1 – 6
Comment 22
Table 1-6: Please indicate the number of samples (n) analysed in footnotes.
Answer
In footnotes added:
n=16/genotype - under Tables 1-3 and n=20/genotype under tables 4-6
Comment 23
LN167: Please indicate the P value in parenthesis (p<0.05)
Answer
added
Comment 24
LN169: Should be revised as: statistically significant....Please indicate the P value in parenthesis (p<0.05)
Answer
added
Comment 25
LN173: Reduce the space.
Answer
deleted
Comment 26
LN174: Please indicate the P value in parenthesis (p<0.05)
Answer
added
Comment 27
LN177: Please indicate the P value in parenthesis (p<0.05)
Answer
added
Comment 28
LN179: Please indicate the P value in parenthesis (p<0.05)
Answer
added
Comment 29
LN180-181: Please rephrase this sentence and indicate the P value in parenthesis (p<0.05)
Answer
added, pigeonpea changed to pigeon
Comment 30
LN184-190: Please indicate the P values in parenthesis where relevant.
Answer
Done.
Comment 31
LN198-207: Please present P values in parenthesis where relevant.
Answer
Added ‘p’ values in parenthesis
Comment 32
LN205: Please delete.
Answer
deleted
Comment 33
LN227: Please mention as: 4.6-6.6%.
Answer
changed
Comment 34
LN274: Please delete.
Answer
deleted
Comment 35
LN282: Please delete.
Answer
deleted „where pectoral muscles play a major role in locomotion”
Comment 36
LN315: Please correct as: carrier pigeon...
Answer
The title of the article is correct. Must be pigeon
Comment 37
LN328: Please revise as: pigeon...
Answer
The title is correct
Comment 38
LN40: Title should be in sentence case. Scientific names must be kept italic.
Answer
Done.
Round 2
Reviewer 2 Report
Comments and Suggestions for Authors
The manuscript has been greatly revised, and I agree to publish it.